# Hall effect in charged conducting ferroelectric domain walls

M.P. Campbell[1], J.P.V. McConville[1], R.G.P. McQuaid[1], D. Prabhakaran[2], A. Kumar[1] & J.M. Gregg[1]

Enhanced conductivity at specific domain walls in ferroelectrics is now an established phenomenon. Surprisingly, however, little is known about the most fundamental aspects of conduction. Carrier types, densities and mobilities have not been determined and transport mechanisms are still a matter of guesswork. Here we demonstrate that intermittent-contact atomic force microscopy (AFM) can detect the Hall effect in conducting domain walls. Studying $YbMnO_3$ single crystals, we have confirmed that p-type conduction occurs in tail-to-tail charged domain walls. By calibration of the AFM signal, an upper estimate of $\sim 1 \times 10^{16}$ cm$^{-3}$ is calculated for the mobile carrier density in the wall, around four orders of magnitude below that required for complete screening of the polar discontinuity. A carrier mobility of $\sim 50$ cm$^2$V$^{-1}$s$^{-1}$ is calculated, about an order of magnitude below equivalent carrier mobilities in p-type silicon, but sufficiently high to preclude carrier-lattice coupling associated with small polarons.

[1] Centre for Nanostructured Media, School of Mathematics and Physics, Queen's University Belfast, Belfast, North Ireland BT71NN, UK. [2] Department of Physics, Clarendon Laboratory, Parks Road, Oxford OX13PU, UK. Correspondence and requests for materials should be addressed to J.M.G. (email: m.gregg@qub.ac.uk).

Domain walls in ferroelectrics are interface structures that separate volumes of differently oriented electrical polarization. As transition regions, their structural, functional and transport properties have long been expected to differ from domain interiors[1–3]. Nevertheless, Seidel et al.'s[4] initial conducting atomic force microscopy (c-AFM) images, of dramatically enhanced conductivity at specific domain walls in BiFeO$_3$, catalysed researchers to see domain walls afresh, as distinct pseudo two-dimensional functional materials. Domain wall conduction has now been seen in numerous ferroelectric systems (for example, BiFeO$_3$ (refs 4–6), Pb(Zr,Ti)O$_3$ (ref. 7), ErMnO$_3$ (ref. 8), LiNbO$_3$ (ref. 9), BaTiO$_3$ (ref. 10), (Ca,Sr)$_3$Ti$_2$O$_7$ (ref. 11), Cu$_3$B$_7$O$_{13}$Cl (R.G.P. McQuaid, private communication), (Bi$_{0.9}$La$_{0.1}$)FeO$_3$ (ref. 12)) and initial concerns that currents were not necessarily genuinely associated with enhanced conductivity (instead arising from undetected domain wall movement for example) have now been firmly allayed. Since the domains themselves are comparatively insulating, domain walls represent isolated conducting channels, which confine currents into narrow sheets. Uniquely, applied fields can cause domain walls to be created or destroyed[13,14], or moved from one point to another[15–17]. Such walls and their associated conduction can, therefore, be actively and dynamically deployed to make mobile nanoscale electrical connections, opening up the potential for novel domain-wall-based electronic devices[18].

Unfortunately, the mechanisms associated with charge transport are not yet clear. There is an expectation that domain walls should have electronic band-structures that differ from domains[19,20], but there is also clear evidence that stress and/or electrostatic interactions with charged point defects can be important[5,8–12,21–23]. Key carrier information is still lacking, as basic transport characterization studies within the walls have been limited. Specifically, carrier types, densities, mobilities and effective masses have not yet been determined. Undoubtedly, experimental difficulties in dealing with domain walls in defined electrical test geometries have been partly to blame, but such problems are not insurmountable, as will be discussed herein.

Several studies on domain wall conduction have noted the importance of polar discontinuities in determining or enhancing conduction[5,8–12]. The established model is that free charges in the ferroelectric matrix migrate to screen the near fields associated with polar discontinuities at 'charged' domain walls[24,25]. Abrupt changes in polarization result in near fields, associated with local distortion in the electronic band structure, which can attract free carriers of opposite polarity, if present, from the bulk material. It is expected that these free carriers both screen near fields and facilitate domain wall conduction. Supporting evidence has been seen in ErMnO$_3$, for example, which is known to have acceptor dopant states in bulk and where conduction is only seen at tail-to-tail walls[8]. Indeed, targeted doping to increase the number density of acceptor states significantly increases conduction[26].

Here we perform local measurement of the Hall effect on mobile charge carriers in domain walls using a novel scanning probe technique. We validate that conduction in tail-to-tail walls is associated with p-type carriers and use quantitative values of the Hall voltage to give an estimate of the carrier mobility and an upper estimate of carrier density.

## Results

### Domain wall structure and characterization.
Our experiments were performed on single crystals of YbMnO$_3$. As with other members of the hexagonal rare earth manganite family, YbMnO$_3$ is an improper ferroelectric. Structural collapse from the high symmetry, high temperature state at a temperature somewhat above 1,250 K (refs 27,28) (termed trimerization) involves coherent buckling of layers of corner-sharing MnO$_5$ trigonal bipyramids that are intercalated with sheets of Yb$^{3+}$ cations. Lengthening and shortening of different O$^{2-}$-Yb$^{3+}$ bonds on cooling, that had been identical in the high-temperature state, cause a redistribution of charge, such that a spontaneous polarization ($\sim 5.6\,\mu C\,cm^{-2}$ at room temperature[29]) results (see references 30,31 for discussions on the phase transition behaviour in the analogous YMnO$_3$ system). The combination of the two equivalent ferroelectric states with dipoles of opposite orientations along the [001] direction and three equivalent senses of structural distortion designated as $\alpha$, $\beta$ and $\gamma$ leads to six discrete, but energetically equivalent possible domains ($\alpha^+$, $\alpha^-$, $\beta^+$, $\beta^-$, $\gamma^+$, $\gamma^-$). Domain walls are not crystallographically confined and so meander quite strongly before converging at characteristic sixfold junctions, as imaged in Fig. 1. Importantly, meandering causes large variations in the orientation of the polarization with respect to the domain walls: walls can, therefore, readily be found across which distinct polarization discontinuities occur. If two sets of positive bound charge meet at such a discontinuity, it is termed 'head-to-head', while two sets of negative bound charge constitute a 'tail-to-tail' boundary. In an analogous rare-earth manganite (ErMnO$_3$ (ref. 8)), enhanced domain wall conductivity is generally found in domain wall sections with tail-to-tail polar discontinuities. Correlation between information on polar orientations of domains obtained by piezoresponse force microscopy (PFM) and local conductivity variations obtained using c-AFM confirms that this is also the case in the YbMnO$_3$ single crystals involved in the current study (Fig. 1).

### Measurement of Hall potential.
For measurement of the Hall effect, gold electrodes were sputtered onto polished surfaces of the YbMnO$_3$ crystals; they were connected to an external DC power supply and a current was driven along percolating conductive domain wall pathways within the interelectrode gap. An electromagnet was lowered into position to provide a magnetic field perpendicular to the applied electric field and this generated a Lorentz force on moving charge carriers (schematic in Fig. 2). The unidirectional displacement of carriers, and consequent charge accumulation at the top surface of the YbMnO$_3$ was then detected using intermittent-contact scanning probe microscopy (tapping mode imaging). The local development of the Hall voltage on conducting domain walls was manifested in the measured topographic image.

Figure 3 shows representative results. A region of YbMnO$_3$ crystal within the interelectrode gap, between the sputtered gold electrodes, was first mapped using standard c-AFM. As expected (from data already obtained in Fig. 1, for example), domain walls at which tail-to-tail polar discontinuities occurred were found to strongly conduct. A series of images of the same area was then taken, monitoring the magnitude and sense of the tip-surface interaction when lateral current (up to 22 $\mu A$) and perpendicular magnetic fields (up to 0.3 T) were applied either separately or at the same time. As can be seen from the data (Fig. 3c), when applied separately, the scanning probe map detects only the scratches in topography that are imaged even in the absence of any applied fields. However, when applied simultaneously, additional contrast was observed. This additional contrast correlated extremely strongly with the c-AFM images of the conducting domain walls in the system. While further sources of contrast cannot be entirely excluded, the requirement for simultaneous magnetic and electric fields is indicative of a Lorentz Force-mediated effect. As such, we attribute the additional contrast solely to the establishment of a Hall potential within the domain

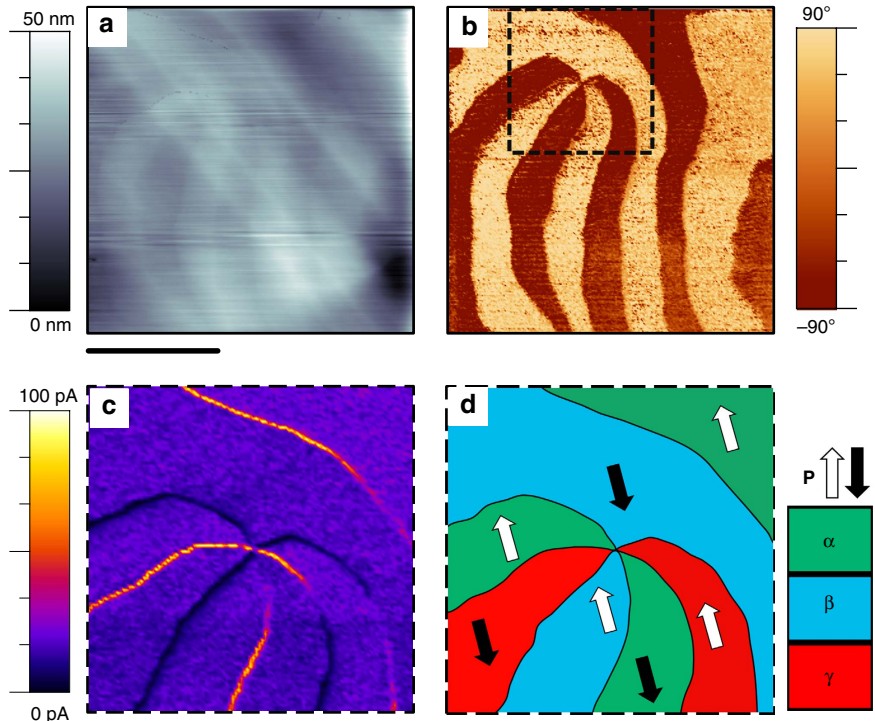

**Figure 1 | Piezoresponse and conductive analysis of domain structure in YbMnO₃.** (**a**) Topographic map of a $5 \times 5\,\mu m^2$ region on an unpolished YbMnO₃ single crystal measured via contact atomic force microscopy. (**b**) Piezoresponse force microscopy map (phase information from lateral mode imaging; cantilever axis horizontal with respect to the image) and (**c**) conductive atomic force microscopy map, taken at 2.3 V, of the subset area showing conduction anomalies along domain walls. (**d**) Structural schematic of the sixfold-vertex region. Taking a circular path around the central vertex it is seen that both the structural and corresponding polarization states alternate, so that no two equivalent structural or ferroelectric domains share a wall. Comparison of the polarization directions implied from **b** (shown in the schematic in **d**) with the current map in **c** confirms that largest currents are measured at tail-to-tail walls. Scale bar for **a,b** is 2 μm and for **c,d** is 1 μm.

wall. Note that Fig. 3b shows the derivative of the topography, as this enhances the contrast associated with the Hall potential at the domain walls (straightforward topography is shown in Supplementary Fig. 1).

**Calibration of Hall potential.** To determine both the sign and magnitude of the Hall voltage developed at the domain walls, a calibration experiment was performed: part of the surface of a {100} polished SrTiO₃ single crystal was sputter-coated with a thin film of gold; this was electrically connected to an external power source. Tapping mode imaging of both the gold film and the adjacent uncoated SrTiO₃ surface was performed under an applied DC voltage of 50 V. This was used to simulate the general background potential of the YbMnO₃ crystal surface in the middle of the interelectrode gap, as illustrated in Fig. 4a (a potential difference between source and drain electrodes of ~100 V was needed to drive the 22 μA lateral current associated with images in Fig. 3). A series of additional voltage pulses with varying magnitude was then superposed onto the DC background and the associated change in the measured 'topography' signal on the gold film and adjacent SrTiO₃ surface monitored. Square voltage pulses were timed such that they occurred at the same spatial point in each sequential AFM scan-line, producing a topographic ramp as can be seen in Fig. 4b. Under these conditions, there appeared to be a reasonably linear relationship between the magnitude of the applied pulse and the measured topographic signal, such that a drop in topography at the domain wall would indicate a negative Hall voltage, whereas an increase in topographic height would indicate a positive Hall voltage (Fig. 4c). We note that equivalent measurements in the absence of

the DC background show an approximately quadratic relationship such that positive and negative Hall signals would be difficult to distinguish (see Supplementary Fig. 2).

Topographic images obtained on the YbMnO₃ domain walls, when current was driven with a perpendicular magnetic field applied, showed that the Hall voltage provided an additional topographic deflection ~1 nm above background (Fig. 4d). Using the calibration described above, this indicates a positive Hall voltage of ~30 mV (see Fig. 4c). The positive nature of the signal is consistent with p-type carriers (see Supplementary Fig. 3). We were not able to distinguish between hole conduction and positive ion-mediated conduction. However, it has been shown[32] that bulk conductivity in YbMnO₃ is mediated by holes and that the ionic contribution is both minor and accounted for by negatively charged cation vacancies. Therefore, given that conduction is only observed at tail-to-tail walls, the large abundance of freely available holes would be most likely to facilitate screening (or partial screening) of the polar discontinuity. Meier *et al.*[8] have made similar suggestions for ErMnO₃ domain wall conduction.

**Calculation of carrier density and mobility.** Using the simplest approximations, the carrier density ($n$) in a domain wall can be related to the Hall voltage ($V_H$) as follows:

$$n = \frac{|\mathbf{I}||\mathbf{B}|}{V_H e d} \qquad (1)$$

where $|\mathbf{I}|$ is the current driven along the domain wall, $|\mathbf{B}|$ is the perpendicular magnetic field, $e$ is the carrier charge and $d$ is the width of the carrier channel (domain wall width). Given that

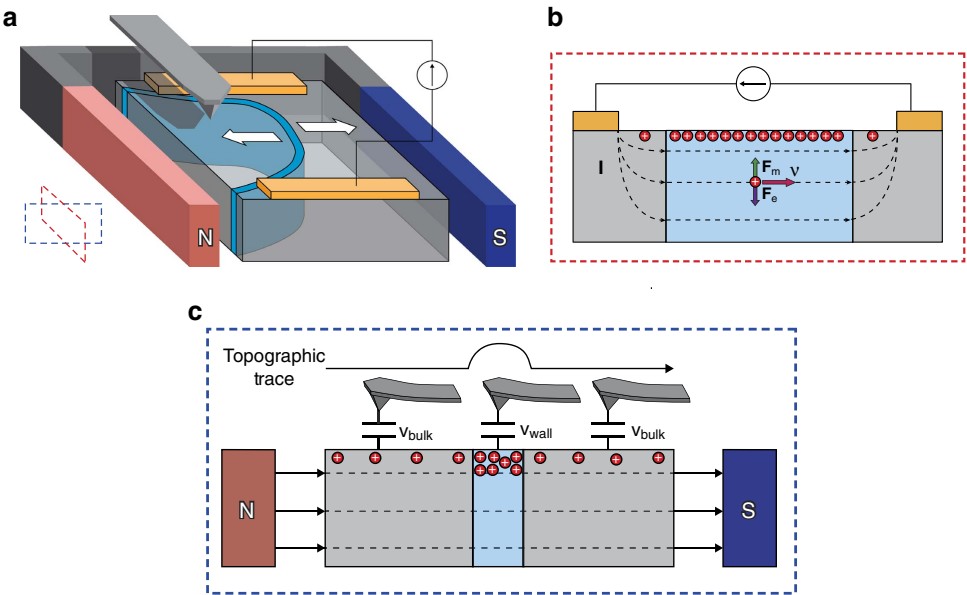

**Figure 2 | Schematic of experimental set-up and operation of Hall voltage imaging. (a)** Schematic of sample geometry. Gold surface electrodes sputtered on YbMnO$_3$ top surface, trapping a tail-to-tail domain wall, shown in blue. A constant current was driven through the surface electrodes and an in-plane magnetic field was applied to the sample, perpendicular to the driven current. Inset: red and blue dashed boxes indicate the relative orientations of the cross-sections shown in **b,c. (b)** Depiction of the Hall effect on a positive test charge, where symbols are as follows: carrier drift velocity $\boldsymbol{\nu}$, Lorentz force $\mathbf{F}_m$, Columbic force $\mathbf{F}_e$, Current along the wall $\mathbf{I}$. **(c)** An atomic force microscopy probe is rastered across the sample surface in tapping mode. It acts as an earthed electrode in a capacitor structure, sensing spatial changes in force-gradients associated with variations in the potential difference between tip and sample surface ($V_{bulk}$ within domains and $V_{wall}$ at the domain wall). The Hall potential developed causes $V_{wall}$ to be distinct from $V_{bulk}$, allowing domain wall contrast to appear in the topographic image.

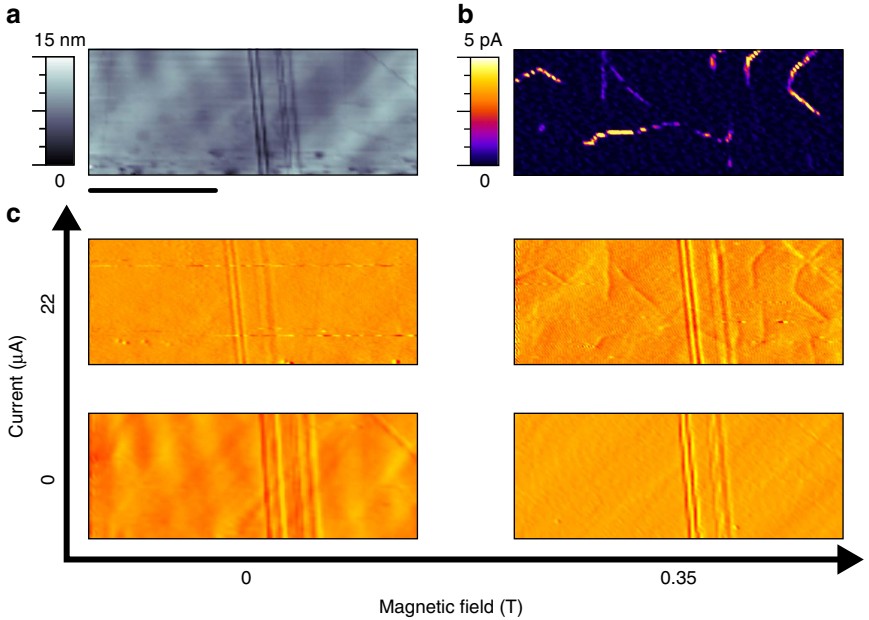

**Figure 3 | Microscopy map of Hall voltage at conducting domain walls. (a)** Contact mode atomic force microscopy (AFM) topography map and **(b)** associated current map within the interelectrode gap, showing a distinct pattern of conducting domain walls without any obvious topographic contrast. **(c)** Derivative of tapping mode AFM topography images taken under different conditions of magnetic field and source-drain current. Domain wall signals are only observed when both lateral current and a transverse magnetic field are applied, suggesting that the additional domain wall contrast is Lorentz-force mediated. Comparison with current map, **b**, confirms strong spatial correspondence. Scale bar, 5 μm.

$V_H \sim 30$ mV, $|\mathbf{B}| \sim 0.3$T, assuming that the carriers are singly charged and that the current of $|\mathbf{I}| \sim 22$ μA is shared among $\sim 10$ domain walls acting in parallel, each of width $\sim 10$ nm, a carrier density of order $n \sim 1 \times 10^{16}$ cm$^{-3}$ can be obtained. Since current is not entirely restricted to the domain walls, this is likely to be a significant overestimate.

The charge density required to screen the polarization discontinuity ($n_{screen}$) found at tail-to-tail walls in YbMnO$_3$ can be estimated by:

$$n_{screen} = \frac{2|\mathbf{P}_S|}{ed} \tag{2}$$

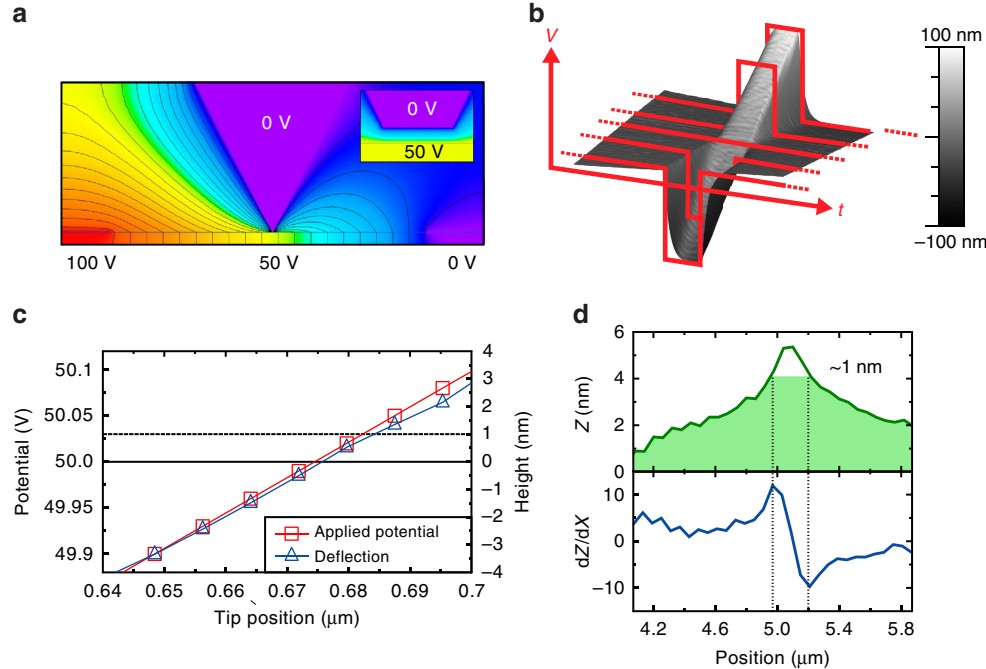

**Figure 4 | Calibration to obtain quantitative estimate for domain wall Hall voltage.** (**a**) Quickfield modelling of the tip and YbMnO₃ surface, with ∼100 V potential difference between source and drain electrodes, verified that measurements of the Hall voltage were set against a DC potential background of around 50 V. (**b**) To calibrate the tip deflection associated with the Hall voltage measurements at domain walls, the topography of a gold-coated SrTiO₃ single crystal was monitored as negative and positive voltage pulses (schematically represented as red lines of V(t)) were supplied to the crystal surface (schematically superimposed onto a background DC potential of 50 V). A smooth correlation between the applied voltage pulse and the topographic deflection was found. (**c**) A section of the measured topography associated with the ramp in **b** and the voltages responsible (voltage pulses plus 50 V DC background). (**d**) The topography signal, Z, across a conducting domain wall in which a Hall voltage has developed and the associated derivative, dZ/dX, show that the above background topographic peak is ∼1 nm in height. This corresponds to a voltage of around 30 mV from the calibration graph in **c** (dashed line).

where $|\mathbf{P}_S|$ is the spontaneous polarization. Taking this to be 5.6 μCcm$^{-2}$ (and again assuming singly charged carriers and an effective domain wall width of 10nm) gives a value for $n_{\text{screen}} \sim 1 \times 10^{20}$cm$^{-3}$, consistent with estimates for BaTiO₃ and PbTiO₃ made previously[24]. Hence, given our upper estimate for the carrier density associated with the measured Hall voltage, if the domain walls are indeed fully screened, only one in every ten thousand screening charges is able to take part in conduction.

The effective carrier mobility can be estimated by re-expressing the current as:

$$|\mathbf{I}| = \frac{V}{R} = |\mathbf{E}|\sigma A \tag{3}$$

where $V$ is the potential difference between source and drain electrodes and $R$ is approximated to be the resistance of the percolating domain walls; $|\mathbf{E}|$ is the estimated uniform electric field in the interelectrode gap, $\sigma$ is the conductivity of the walls and $A$ is the effective cross-sectional area of the walls. Using the relationship:

$$\sigma = ne\mu \tag{4}$$

where $\mu$ is the carrier mobility, the Hall voltage can be expressed as:

$$V_H = \frac{|\mathbf{E}|ne\mu A|\mathbf{B}|}{ned} = \frac{|\mathbf{E}|\mu A|\mathbf{B}|}{d} \tag{5}$$

and taking the cross-sectional area of the domain wall as the width multiplied by the depth from the top surface (D), the carrier mobility can then be given as:

$$\mu = \frac{V_H}{|\mathbf{E}|D|\mathbf{B}|} \tag{6}$$

Given a Hall voltage of ∼30 mV, an effective electric field of $1 \times 10^7$ Vm$^{-1}$, a magnetic field of 0.3 T and an estimated effective wall depth as being on the order of the coarseness of the domain microstructure evident at the top surface (a few microns), p-type carrier mobilities of ∼50 cm$^2$V$^{-1}$s$^{-1}$ can be estimated. This is about an order of magnitude lower than that typical of p-type silicon with a similar carrier density[33]. However, it is several orders of magnitude higher than that typical of small polarons in oxides[34,35] and at the high end of that typical of large polarons[36]. It is, therefore, possible that transport along the walls does not involve significant lattice coupling, but categorical statements will require further measurements.

## Discussion

A novel scanning probe microscopy technique has been demonstrated to allow measurement of the Hall effect, with spatial resolution on the order of tens of nanometers. Measurements on conducting domain walls in YbMnO₃ reveal that current is due to p-type carriers, which in this case are very likely to be holes. In addition, quantitative estimates for carrier mobilities and densities have been made. Further investigation is under way to fully elucidate the contrast mechanism and facilitate greater precision in measurement of the Hall potential. Nonetheless, our measurements represent the first explicit characterization of the carriers involved in conducting ferroelectric domain walls.

## Method

**YbMnO₃ crystal growth.** Stoichiometric amounts of high purity (>99.99%) Yb₂O₃ and MnO₂ powders were mixed and calcined in air at 1,100 °C and 1,200 °C for 36 h each with intermediate grinding. The reacted powder was then formed into

cylindrical feed rod shapes, 10 mm in diameter and 10 cm in length, and sintered at 1,250 °C for 24 h in air. Single crystals of YbMnO$_3$ were then grown using an optical floating-zone technique (Crystal Systems Inc). The growth was carried out at a rate of 2–3 mm h$^{-1}$ in a flow (200 cc min$^{-1}$) of Ar/O$_2$ mixed gas atmosphere with a feed and seed rod rotation at 20 r.p.m.

**PFM and c-AFM mapping.** PFM measurements were carried out with a Veeco Dimension 3100 AFM system, modified for PFM, with a Nanoscope IIIa controller using a EG&G 7256 lock-in amplifier. Here an AC bias of 5 V$_{RMS}$ with a frequency of 20 kHz was applied to the base of the sample via a silver paste bottom electrode. Current maps were obtained on the same system using an additional Bruker Tunnelling AFM (TUNA) module. Here a DC bias was applied to a silver paste bottom electrode and a lateral gold electrode, and the tip grounded, for data in Figs 1 and 3, respectively. For all measurements platinum/iridium coated silicon probes, Nanosensors model PPP-EFM, were used with a force constant ~2.8 Nm$^{-1}$.

**Hall effect imaging.** Hall measurements were carried out with a Bruker Dimension 5000, modified to include a reversible electromagnet with field strengths of up to 0.4 T, in conjunction with a Nanoscope V controller. Nanosensor model PPP-EFM probes were used. A constant DC current of up to 22 μA was driven across the interelectrode gap, orthogonal to the applied magnetic field, using a Keithley 6221 current source. Tapping mode topography scans were carried out, at a scan rate of 0.1 Hz, at a range of applied currents and magnetic field strengths. The resulting interaction between the probe and established Hall potential led to deflection of the tip which was monitored with nanoscale spatial resolution such that domain wall-specific signals could be identified.

**Data availability.** The data supporting the findings of this study are available from the corresponding author upon request.

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

## Acknowledgements

We acknowledge funding support from the Engineering and Physical Sciences Research Council (EPSRC) through contracts EP/J017825 and EP/N018389, equipment donation from Seagate Technology (Bruker Dimension 5000 with *in-situ* magnetic field capability) and studentship support through the Northern Ireland Department for Employment and Learning. We also thank Prof Roger Whatmore for introducing the Oxford and Belfast research teams to each other.

## Author contributions

The scanning probe microscopy and specific experiment design modifications needed for the Hall voltage measurements were performed by M.P.C., R.G.P.McQ. and A.K. YbMnO$_3$ single crystals were grown by D.P.. Calibration measurements and analysis were performed by J.P.V.McC. and M.P.C. The idea for the experiment was conceived by J.M.G. who supervised the research in collaboration with A.K. All authors were involved in the manuscript preparation.

## Additional information

**Competing financial interests**: The authors declare no competing financial interests.

**How to cite this article**: Campbell M. P. *et al.* Hall effect in charged conducting ferroelectric domain walls. *Nat. Commun.* **7**, 13764 doi: 10.1038/ncomms13764 (2016).

**Publisher's note**: 

