## [Peer Review File · Nature Communications]

Reviewers' comments:

Reviewer #1 (Remarks to the Author):

Campbell and co-authors present Hall effect measurements at conducting ferroelectric domain walls in YbMnO₃ which are realized by adapting intermittent-contact atomic force microscopy (AFM). AFM measurements performed under different applied electric and magnetic fields detect a domain-wall-related signal of electrostatic origin, providing direct evidence for the proposed p-type conductance at tail-to-tail domain walls in hexagonal manganites. Up to now, the p-type nature of the tail-to-tail domain walls in hexagonal manganites has been shown only indirectly. The key result of the present work is the demonstration of Hall measurements at the local scale. Here, the authors go beyond previous studies and succeed in identifying a powerful tool that is of strong interest for the research on domain wall and nanoscience in general. The results are well presented and convincing. However, taken into account that the main novelty of this work lies in the introduction of a new AFM technique, it would be of interest to see a more quantitative analysis and / or additional experiments that demonstrate the potential scope of application and wider impact of this method. In particular, it would be of interest to learn about the capability to evaluate the carrier densities, mobilities, and effective mass which the authors address in their introduction. I believe that this work bears great potential, but I am not really convinced that it should be published in Nature Communications in its present form. I strongly encourage a resubmission in the future though.

Besides the above remarks, I suggest to consider the following points during revision:

1. The authors write that the carrier type, densities, mobilities and effective mass are still a matter of guesswork and call the accumulation of mobile carriers at charged domain walls a working hypothesis. I agree that densities, mobilities and effective mass are yet to be measured and it would be great to see some quantitative results based on the Hall measurements presented in this work. There is an increasing amount of literature, however, calculating the electric properties at charged domain walls (see, e.g., Phys. Rev. B 83, 235313 and Phys. Rev. B 83, 184104 (2011)) and experimental evidence for the p-type nature of carriers is available - at least indirectly - from synthesis-related studies (Adv. Electron. Materials 2, 1500195 (2015)). Thus, I would suggest to revise the manuscript accordingly to avoid the impression that nothing is known about the transport mechanism. The method presented in the work is fascinating by itself and does not need an additional selling point, especially because it does not provide new quantitative insight concerning densities, mobilities and effective mass at this point. In the introduction the authors further write "...transport properties have long been expected to differ from domain interiors". Here, it might be nice to cite some of the earlier literature on charged ferroelectric walls, e.g., Ferroelectrics 6, 29-31 (1973). The idea of conducting charged walls is around for quite some time and a growing number of theoretical studies render the redistribution of carriers and screening-based effects more than a "hypothesis" (Sci. Rep. 5, 15819 (2015)).

2. In some sections the use of more technical phrases may be appropriate. For instance, the authors are talking about a "structural collapse" concerning the high-temperature transition in YbMnO₃. In addition to reference [23], key studies on the nature of the transition (Nature Mater. 3, 164-170 (2004), Nature Phys. 11, 1070-1073 (2015)) and transition temperature in YbMnO₃ (Phys. Rev. Lett. 108, 167603 (2012)) may be cited. Another example are the "set of positive poles" at head-to-head walls, which are conventionally termed polarization charges or bound charges.

3. It would be nice to present some more details about the sample growth in the Methods section.

4. Given the fact that only a quarter of the manuscript is about new results, i.e., local Hall data, I would suggest to present all data in the main text instead of "hiding" some of it in the Supplementary

Information. Moreover, a stronger emphasis should be put on the experimental details. Are quantitative measurements possible, what are the limitations, and can we expect to learn more about densities and carrier mobilities in the future?

Reviewer #2 (Remarks to the Author):

The manuscript presents the Hall effect measured with a scanning probe tip on charged domain walls in an improper ferroelectric manganate.

The text is pleasure to read, because it is a very simple, straightforward presentation of an ingenious method which confirms the expected p-type conductivity at tail-to-tail domain walls. The presented data are novel, credible, qualitatively robust, and the claims and references are in my opinion appropriate. Therefore, I recommend the manuscript for publication in Nature Communications already in the current form.

I however suggest the authors to consider refinement of one issue in the text: The theory behind the free-charge accumulation at charged domain walls is a bit more complicated then it might seem to be from the presented verbal explanation. In the manuscript it sounds as if the free charge was accumulated at a charged domain wall in the same way it is accumulated on a charged plane in a free space. Free carriers in crystals need to be in a conduction band and under the Fermi level. This requires some necessary band bending. Indeed, this bending should be measurable as potential jumps at charged domain walls. It is therefore rather interesting to see that no potential relief is seen at charged domain walls in the absence of electric and magnetic fields.

Reviewer #3 (Remarks to the Author):

Campbell et al. describe a novel method to characterize domain walls based on the Hall effect that acts on charge carriers within the domain walls in ferroelectric materials. The idea and presentation are original and interesting to the community and therefore should potentially be considered for publication.

The Hall effect as such is a standard characterization method for semiconductor materials that gives information about carrier type, carrier concentration and mobility. Currently, the manuscript only discusses carrier type and shows that hole conduction is present in the investigated hexagonal manganite material, which was already found by other groups and published.

Therefore, in order for this manuscript to be considered further and show that this indeed is an innovative and new way of characterizing domain walls, I believe that the other two points, i.e. carrier density and mobility should be investigated closer with this method and the obtained results should be presented and discussed within this manuscript.

REVIEWERS' COMMENTS:

Reviewer #1 (Remarks to the Author):

The authors responded to all questions raised by the reviewers in a satisfactory way. Most importantly, they succeeded in realizing the requested challenging calibration experiments, which now allow for quantitative Hall measurements at the nanoscale. Here, the team did a really great job and pushed the paper to a whole new level, so that the outcome of this work will be of great value for the community and also beyond. In conclusion, I fully support publication of the revised manuscript in Nature Communications.

Reviewer #2 (Remarks to the Author):

The modified version of the manuscript deals, in my opinion, satisfactorily with the issues related to the quantitative estimates raised by other reviewers. I agree that it increased the volume of useful information in this work and that it represents an improvement. Since I was satisfied already with the previous version only with qualitative results, I again recommend the manuscript for publication in its current form.

Reviewer #3 (Remarks to the Author):

The authors have made an effort to quantify mobility and carrier density within their measurement and satisfactorily answered previous questions. The manuscript is now in publishable form.

Referee 1:

Comment 1.1: *“it would be of interest to see a more quantitative analysis and / or additional experiments that demonstrate the potential scope of application and wider impact of this method. In particular, it would be of interested to learn about the capability to evaluate the carrier densities, mobilities, and effective mass which the authors address in their introduction” and “The authors write that the carrier type, densities, mobilities and effective mass are still a matter of guesswork and call the accumulation of mobile carriers at charged domain walls a working hypothesis. I agree that densities, mobilities and effective mass are yet to be measured and it would be great to see some quantitative results based on the Hall measurements presented in this work.”*

Response 1.1: To obtain values for the carrier density and mobility (in addition to the carrier type), we had to quantify the Hall Voltage developed at the conducting domain walls. To do this, we calibrated the system: part of the surface of a polished SrTiO₃ single crystal was sputter-coated with gold and electrically connected to an external power source / function generator. The tapping-mode topography of the gold surface (and the SrTiO₃ surface adjacent to the gold) was then monitored under different states of electrical bias. Since the initial measurements were taken in the centre of the interelectrode gap between the source and drain electrodes and the current in the domain walls was driven using ~100V potential difference, we calibrated the topographic signal against a background dc voltage of 50V. A series of square pulses was superposed onto this 50V dc background and the effect that these additional pulses had on the “topography” signal was recorded. In the actual imaging experiments the Hall Voltage at the domain walls induced ~1nm of additional “topography” over and above that evident without current, magnetic fields and the associated Lorenz force on carriers. Using our calibration data, we were able to equate this to a Hall Voltage of the order of 30mV.

With this value and estimates for domain wall width, applied field etc we were able to make numerical estimates for the maximum active carrier density in the walls (~1 x 10¹⁶ cm⁻³) and approximate mobility of the carriers (~50 cm²V⁻¹s⁻¹). The carrier density is around 4 orders of magnitude below that which would be needed to electrically screen the polarisation discontinuity, which suggests either the walls are not fully screened, or that free carriers are a minor component in the screening mechanism (perhaps immobile defect states facilitate the bulk of the screening). Carrier mobilities are quite large for oxides, but an order of magnitude below that typical of p-type silicon. We note, however, that the mobility determined precludes small polarons as the active carrier they have maximum mobilities in oxides of ~0.1cm²V⁻¹s⁻¹. It is also on the high end of mobilities found for large polarons and so we suggest that conduction in the walls might not involve significant lattice coupling.

Changes to Manuscript: A new section of text has been added, describing the calibration technique, the results obtained and the implications for carrier densities and mobilities. An additional figure has been added. Text and figure are as follows:

“To determine both the sign and magnitude of the Hall Voltage developed at the domain walls, a calibration experiment was performed: part of the surface of a {100} polished SrTiO₃ single crystal was sputter-coated with a thin film of gold; this was electrically connected to an external power source. Tapping mode imaging of both the gold film and

the adjacent uncoated SrTiO₃ surface was performed under an applied dc voltage of 50V. This was used to simulate the general background potential of the YbMnO₃ crystal surface in the middle of the interelectrode gap, as illustrated in Figure 4a (a potential difference between source and drain electrodes of ~100V was needed to drive the 22μA lateral current associated with images in Figure 3). A series of additional voltage pulses with varying magnitude was then superposed onto the dc background and the associated change in the measured “topography” signal on the gold film and adjacent SrTiO₃ surface monitored. Square voltage pulses were timed such that they occurred at the same spatial point in each sequential AFM scan-line, producing a topographic ramp as can be seen in Figure 4b. Under these conditions, there appeared to be a reasonably linear relationship between the magnitude of the applied pulse and the measured topographic signal, such that a drop in topography at the domain wall would indicate a negative Hall Voltage, whereas an increase in topographic height would indicate a positive Hall Voltage (Figure 4c). We note that equivalent measurements in the absence of the dc background show an approximately quadratic relationship such that positive and negative Hall signals would be difficult to distinguish (see Figure S2 in the supplementary materials).

Topographic images obtained on the YbMnO₃ domain walls, when current was driven with a perpendicular magnetic field applied, showed that the Hall Voltage provided an additional topographic deflection ~1nm above background (Figure 4d). Using the calibration described above, this indicates a positive Hall Voltage of ~30mV (see Figure 4c). The positive nature of the signal is consistent with p-type carriers (see Figure S3 in the supplementary materials). We were not able to distinguish between hole conduction and positive ion-mediated conduction. However, it has been shown [33] that bulk conductivity in YbMnO₃ is mediated by holes and that the ionic contribution is both minor and accounted for by negatively charged cation vacancies. Therefore, given that conduction is only observed at tail-to-tail walls, the large abundance of freely available holes would be most likely to facilitate screening (or partial screening) of the polar discontinuity. Meier *et al.* [8] have made similar suggestions for ErMnO₃ domain wall conduction.

Using the simplest approximations, the carrier density (n) in a domain wall can be related to the Hall Voltage (V_H) as follows:

$$n = \frac{IB}{V_H e d} \quad (1)$$

where I is the current driven along the domain wall, B is the perpendicular magnetic field, e is the carrier charge and d is the width of the carrier channel (domain wall width). Given

that $V_H \sim 30\text{mV}$, $B \sim 0.3\text{T}$, assuming that the carriers are singly charged and that the current of $I \sim 22\mu\text{A}$ is shared among ~ 10 domain walls acting in parallel, each of width $\sim 10\text{nm}$, a carrier density of order $n \sim 1 \times 10^{16} \text{ cm}^{-3}$ can be obtained. Since current is not entirely restricted to the domain walls, this is likely to be a significant overestimate.

The charge density required to screen the polarization discontinuity (n_{screen}) found at tail-to-tail walls in YbMnO_3 can be estimated by:

$$n_{\text{screen}} = 2P_s / ed \quad (2)$$

where P_s is the spontaneous polarization. Taking this to be $5.6\mu\text{C cm}^{-2}$ (and again assuming singly charged carriers and an effective domain wall width of 10nm) gives a value for $n_{\text{screen}} \sim 1 \times 10^{20} \text{ cm}^{-3}$, consistent with estimates for BaTiO_3 and PbTiO_3 made previously [25]. Hence, given our upper estimate for the carrier density associated with the measured Hall Voltage, if the domain walls are indeed fully screened, only one in every ten thousand screening charges is able to take part in conduction.

The effective carrier mobility can be estimated by re-expressing the current as:

$$I = \frac{V}{R} = E\sigma A \quad (3)$$

where V is the potential difference between source and drain electrodes and R is approximated to be the resistance of the percolating domain walls; E is the estimated uniform electric field in the interelectrode gap, σ is the conductivity of the walls and A is the effective cross-sectional area of the walls. Using the relationship:

$$\sigma = ne\mu \quad (4)$$

where μ is the carrier mobility, the Hall Voltage can be expressed as:

$$V_H = \frac{Ene\mu AB}{ned} = \frac{E\mu AB}{d} \quad (5)$$

and taking the cross-sectional area of the domain wall as the width multiplied by the depth from the top surface (D), the carrier mobility can then be given as:

$$\mu = \frac{V_H}{EBD} \quad (6)$$

Given a Hall Voltage of $\sim 30 \text{ mV}$, an effective electric field of $1 \times 10^7 \text{ Vm}^{-1}$, a magnetic field of 0.3T and an estimated effective wall depth as being on the order of the coarseness of the domain wall microstructure evident at the top surface (a few microns), p-type carrier mobilities of $\sim 50\text{cm}^2\text{V}^{-1}\text{s}^{-1}$ can be estimated. This is about an order of magnitude lower than that typical of p-type silicon with a similar carrier density [34]. However, it is several orders of magnitude higher than that typical of small polarons in oxides [35, 36] and at the high end of that typical of large polarons [37]. It is therefore possible that transport along

the walls does not involve significant lattice coupling, but categorical statements will require further measurements.

Figure 4

Calibration to obtain quantitative estimate for domain wall Hall Voltage. **a**, Quickfield modelling of the tip and YbMnO₃ surface, with ~100V potential difference between source and drain electrodes, verified that measurements of the Hall Voltage were set against a dc potential background of around 50V. **b**, To calibrate the tip deflection associated with the Hall voltage measurements at domain walls, the topography of a gold-coated SrTiO₃ single crystal was monitored as negative and positive voltage pulses (schematically represented as red lines of $V(t)$) were supplied to the crystal surface (superimposed onto a background dc potential of 50V). A smooth correlation between the applied voltage pulse and the topographic deflection was found. **c**, A section of the measured topography associated with the ramp in b and the voltages responsible (voltage pulse plus 50V dc background). **d**, The topography signal across a conducting domain wall in which a Hall voltage has developed (top panel) and the associated derivative (bottom panel) show that the above-background topographic peak is approximately 1nm in height. This corresponds to a voltage of around 30mV from the calibration graph in c (dashed line)."

NOTE: The approximately linear relationship between the observed deflection in the topography signal and the voltage pulse magnitude which causes it is strongly influenced by the fact that there is a background dc potential. For completeness, we have recorded the topographic deflection without any dc potential and found an approximately quadratic dependence. Clearly, Hall Voltages need currents to be driven between source and drain and hence dc potentials will be a feature of all measurements. Nevertheless, we thought it would be instructive for readers to have access to the zero bias quadratic data and have presented it in the supplementary material.

Comment 1.2: *There is an increasing amount of literature, however, calculating the electric properties at charged domain walls (see, e.g., Phys. Rev. B **83**, 235313 and Phys. Rev. B **83**, 184104 (2011)) and experimental evidence for the p-type nature of carriers is available - at least indirectly - from synthesis-related studies (Adv. Electron. Materials **2**, 1500195 (2015)). Thus, I would suggest to revise the manuscript accordingly to avoid the impression that nothing is known about the transport mechanism. The method presented in the work is fascinating by itself and does not need an additional selling point, especially because it does not provide new quantitative insight concerning densities, mobilities and effective mass at this point. In the introduction the authors further write "...transport properties have long been expected to differ from domain interiors". Here, it might be nice to cite some of the earlier literature on charged ferroelectric walls, e.g., Ferroelectrics **6**, 29-31 (1973). The idea of conducting charged walls is around for quite some time and a growing number of theoretical studies render the redistribution of carriers and screening-based effects more than a "hypothesis" (Sci. Rep. **5**, 15819 (2015)).*

Response 1.2 and changes made: We accept the point made by the referee and have changed the manuscript to better represent previous pertinent research in which transport mechanism theories have been developed and relevant experiments performed. The majority of the suggested references have also been added.

Comment 1.3: *"In some sections the use of more technical phrases may be appropriate. For instance, the authors are talking about a "structural collapse" concerning the high-temperature transition in YbMnO₃. In addition to reference [23], key studies on the nature of the transition (Nature Mater. **3**, 164-170 (2004), Nature Phys. **11**, 1070-1073 (2015)) and transition temperature in YbMnO₃ (Phys. Rev. Lett. **108**, 167603 (2012)) may be cited. Another example are the "set of positive poles" at head-to-head walls, which are conventionally termed polarization charges or bound charges."*

Response 1.3 and changes made: Nomenclature has been tidied and references added.

Comment 1.4: *"It would be nice to present some more details about the sample growth in the Methods section."*

Response 1.4 and changes made: A section has been added in the methods section as follows:

“YbMnO₃ Crystal Growth - Stoichiometric amounts of high purity (>99.99%) Yb₂O₃ and MnO₂ powders were mixed and calcined in air at 1100°C and 1200°C for 36h each with intermediate grinding. The reacted powder was then formed into cylindrical feed rod shapes, 10mm in diameter and 10cm in length, and sintered at 1250°C for 24 h in air. Single crystals of YbMnO₃ were then grown using an optical floating-zone technique

(Crystal Systems Inc). The growth was carried out at a rate of 2-3 mm/h in a flow (200cc/min) of Ar/O₂ mixed gas atmosphere with a feed and seed rod rotation at 20 rpm.”

Comment 1.5: *“Given the fact that only a quarter of the manuscript is about new results, i.e., local Hall data, I would suggest to present all data in the main text instead of “hiding” some of it in the Supplementary Information. Moreover, a stronger emphasis should be put on the experimental details. Are quantitative measurements possible, what are the limitations, and can we expect to learn more about densities and carrier mobilities in the future?”*

Response 1.5: We believe that the inclusion of the description of the calibration measurements, the additional figure specifically on calibration and the associated quantitative evaluation of the carrier density and mobility have changed the balance of the manuscript significantly and thank the referees for prompting us to do this work. Given the additional material, we didn’t think it was necessary to include the other data that had been in the supplementary material, as this did not add anything new in terms of physical insight.

Referee 2:

Comment 2.1: *“I however suggest the authors to consider refinement of one issue in the text: The theory behind the free-charge accumulation at charged domain walls is a bit more complicated than it might seem to be from the presented verbal explanation. In the manuscript it sounds as if the free charge was accumulated at a charged domain wall in the same way it is accumulated on a charged plane in a free space. Free carriers in crystals need to be in a conduction band and under the Fermi level. This requires some necessary band bending. Indeed, this bending should be measurable as potential jumps at charged domain walls. It is therefore rather interesting to see that no potential relief is seen at charged domain walls in the absence of electric and magnetic fields.”*

Response 2.1 and changes made: We accept the comment made by the referee and have noted the requirement for localised band bending or at least band distortion in the revised text as follows:

“Abrupt changes in polarization result in near fields, associated with local distortion in the electronic band structure, which can attract free carriers of opposite polarity from the bulk material, if present.”

We have not made comment about seeing the effects of local band-bending in the text, as we think it could confuse the readers as to the origin of the Hall Voltage contrast that we do observe.

Referee 3:

Comment 3.1: *“Currently, the manuscript only discusses carrier type and shows that hole conduction is present in the investigated hexagonal manganite material, which was already found by other groups and published. Therefore, in order for this manuscript to be considered further and show that this indeed is an innovative and new way of characterizing domain walls, I believe that the other two points, i.e. carrier density and*

mobility should be investigated closer with this method and the obtained results should be presented and discussed within this manuscript.”

Response 3.1: We accept the referee's point and refer to “Response 1.1” given above. We are genuinely grateful for the insistence of the referees that we should go further and determine quantitative information on carrier density and mobility. We think that this has dramatically improved the manuscript and made it more useful to the community.

Manuscript NCOMMS-16-11830A (Campbell *et al.*) Response to Referee Comments II

All three referees gave entirely positive reviews on the altered manuscript and so no technical changes have been made (only editorial changes). The authors thank the referees for their original comments as they helped us to improve the article substantially.

Referee 1:

The authors responded to all questions raised by the reviewers in a satisfactory way. Most importantly, they succeeded in realizing the requested challenging calibration experiments, which now allow for quantitative Hall measurements at the nanoscale. Here, the team did a really great job and pushed the paper to a whole new level, so that the outcome of this work will be of great value for the community and also beyond. In conclusion, I fully support publication of the revised manuscript in Nature Communications.

Response 1: No further changes made.

Referee 2:

The modified version of the manuscript deals, in my opinion, satisfactorily with the issues related to the quantitative estimates raised by other reviewers. I agree that it increased the volume of useful information in this work and that it represents an improvement. Since I was satisfied already with the previous version only with qualitative results, I again recommend the manuscript for publication in its current form.

Response 2: No further changes made.

Referee 3:

The authors have made an effort to quantify mobility and carrier density within their measurement and satisfactorily answered previous questions. The manuscript is now in publishable form.

Response 3: No further changes made.